# Background-free $^{12}$C($\alpha$, $\gamma$) angular distribution measurements with a time projection chamber operating in Gamma beams

Kristian C. Z. Haverson [1] ✉, Robin Smith [1,2], Moshe Gai [2], Deran K. Schweitzer[2], Sarah R. Stern [2] & Sean W. Finch [3]

The carbon oxygen ratio (C/O) at the end of stellar helium burning is a crucial nuclear input to stellar evolution theory. Knowledge of the C/O ratio with sufficient accuracy has eluded measurement over the past five decades. It is determined by the rate of oxygen formation in the fusion of helium with $^{12}$C, denoted as $^{12}$C($\alpha$, $\gamma$)$^{16}$O. Even though recent methods employing a time projection chamber can measure the time-reverse photo-dissociation reaction, the results still do not show unambiguous agreement with the predictions of quantum scattering theory. Here, we improve this method using a N$_2$O gas target. This improvement allows us to eliminate the background caused by $^{12}$C photo-dissociation events, obtain complete angular distributions ($0°-180°$), and measure the cross sections over the $1^-$ resonance in $^{16}$O at $E_{cm} \sim 2.4$ MeV. These measurements resolve the discrepancy that was previously observed between the measured $E1-E2$ mixing phase angle ($\phi_{12}$) of $^{12}$C($\alpha$, $\gamma$)$^{16}$O and the predictions of quantum scattering theory. This newfound agreement demonstrates the viability of our method for conducting measurements at lower energies.

Stellar Evolution theory[1], the brainchild of nuclear physics, is a mature theory that has reached, in some cases, sufficient precision to study fundamental physics. Most notable is neutrino astrophysics and the standard model of the sun[2] that led to the first observation of neutrino mass, hence the first deviation from the standard model of particle physics[3]. Stellar Evolution theory describes supernova explosions with great success, as discussed by Bethe and Brown in their Scientific American article[4]. Of particular interest is the final fate of Type II supernovae (SNeII), a neutron star or a black hole.

Supernova 1987A (SN1987A) was the first observed by the naked eye since Kepler's observation on October 8, 1604. The ultimate fate of SN1987A (neutron star or black hole) is important for resolving the fundamental property of the equation of state (EOS) of nuclear matter; a soft EOS predicted the collapse of 1987A to a black hole[5]. Recent observations from the James Webb telescope[6] of blueshifted narrow infrared emission lines of argon and sulphur were explained by the effects of ionising radiation from a neutron star. This is in alignment with the earlier observation of a neutrino burst coinciding with SN1987A, with a duration suggesting the presence of a neutron star[7].

In addition to the EOS, the fate of SNeII depends on the carbon-to-oxygen (C/O) ratio at the end of a star's helium burning. During this burning stage, $^{12}$C is produced in the two steps of the so-called 3$\alpha$ process that was proposed by Fred Hoyle[8], and $^{16}$O is produced in the further fusion of $^{12}$C with an alpha-particle in the $^{12}$C($\alpha$, $\gamma$) reaction. We emphasise that already early on, Hoyle noted[8] that "It can be shown that [the] reaction ($^{12}$C($\alpha$, $\gamma$)) is the most effective in destroying $^{12}$C. Hence, to decide how far $^{12}$C accumulates, it is necessary to compare the rates of the reactions (3$\alpha$ and $^{12}$C($\alpha$, $\gamma$))". Since the 3$\alpha$ rate is currently well constrained[9,10] (±11%), the uncertainty on the C/O ratio at the end of helium burning is directly dependent on the exact value of the cross-section of the $^{12}$C($\alpha$, $\gamma$)$^{16}$O reaction, as first noted by Hoyle[8] (and considerably later by Fowler[11]).

For a progenitor star with a C/O ratio smaller than 1, the oxygen-rich star is predicted to leave behind a black hole[12]. Knowledge of the cross-section of this reaction would allow a prediction of the minimum mass of the progenitor star that could yield a black hole. For this, the value of the cross-section of the $^{12}$C($\alpha$, $\gamma$)$^{16}$O reaction at the stellar energy of 300 keV must be extracted from laboratory measurements with a precision of 5−10%.

[1]School of Engineering & Built Environment, Sheffield Hallam University, Sheffield, UK. [2]Laboratory for Nuclear Science at Avery Point, University of Connecticut, Groton, CT, USA. [3]Department of Physics and Triangle Universities Nuclear Laboratory, Duke University, Durham, NC, USA. ✉e-mail: kristianzajdek@gmail.com

As such, direct measurements of this cross-section have been the ultimate, and indeed the most important goal of nuclear astrophysics over the last five decades[11]. Aside from the fate of SNeII, it affects, among many other issues, the light curves of Type Ia supernovae (SNeIa)[13] and the black hole mass gap[14]. However, despite half a century of experimental efforts to measure the $^{12}C(\alpha, \gamma)$ reaction, large uncertainties remain in the data measured at higher energies. In this paper, we address only the data measured at laboratory energies and do not consider extrapolation to the stellar energy at the Gamow window. Simply put, as we demonstrate below, existing data of the $^{12}C(\alpha, \gamma)$ reaction do not allow for precise extrapolation.

Previous measurements of the cross-section of the $^{12}C(\alpha, \gamma_0)$ reaction used large arrays of gamma detectors to directly measure gamma-ray angular distributions[15–21]. The measured angular distributions are fitted with a partial wave decomposition[15] to extract the $E1$ and $E2$ cross-sections, and their interference angle $\phi_{12}$. It was demonstrated[22,23] that the interference angle can be obtained from the measured elastic scattering phase shifts[24–26] using equation (1)[15,22,23]:

$$\phi_{12} = \delta_2 - \delta_1 + \tan^{-1}(\eta/2), \quad (1)$$

where $\delta_1$ and $\delta_2$ are the measured $p$- and $d$- wave elastic scattering phase shifts, and $\eta$ is the Sommerfeld parameter, $\eta = Z_1 Z_2 \times \alpha/\beta$, where $\alpha$ is the fine structure constant and $\beta = v/c$. Equation (1) was originally derived using $R$-matrix theory[15,22,23]. But the general validity of this relation was established by Knutson[27], and later by Brune[28] and Gai[29] by showing it to be a consequence of the Watson theorem[30], hence unitarity—a prediction of quantum mechanics itself. Such a clear prediction we claimed[31] should be used to test the quality of measured angular distributions, as we discuss here.

A review of current measured data of the $^{12}C(\alpha, \gamma_0)$ reaction[15–21] was presented by Gai[32]. In particular, this review includes a figure summarising measurements of $\phi_{12}$ over the $1^-$ resonance at $\sim 2.4$ MeV (prepared in 2006 by Professor Wolfgang Hammer—former PI of the Stuttgart group), where $\phi_{12}$ varies rapidly. This figure, shown in Supplementary Fig. 1, demonstrates that current data do not agree with the very prediction of quantum mechanics (labelled as $R$-matrix fit).

On this figure, the data of the Stuttgart group quote $\phi_{12}$ values with small error bars, especially at $\sim 2.3$ MeV, where the cross-sections are large[19]. All nine data points below 2.3 MeV, significantly disagree with predicted values from equation (1). In Table I of[19] we note that the extracted $E1/E2$ ratios differ by up to a factor 4 depending on whether the value of $\phi_{12}$ is fixed by equation (1) or considered as a free fit parameter. Furthermore, analyses of the Stuttgart data[29] reveal $E2/E1$ ratios with error bars that are considerably larger than they quoted.

Inspecting more than 100 raw gamma-ray spectra shown in the appendix of Fey's Ph.D. thesis[33], and shown again by Gai[32], reveals no discernible gamma-lines above background at energies below 1.7 MeV. A clear gamma-ray line of the $^{12}C(\alpha, \gamma_0)$ is observed in spectra measured at $E_{cm} = 2.2$ MeV, but the measured $\cos \phi_{12}$ value is a factor of 2 larger than predicted by equation (1). This systematic discrepancy must be understood before such data can be used for extrapolation, an issue that has been overlooked in previous analyses[34]. Furthermore, as pointed out by the authors of ref.[34], this extrapolation essentially relies on the asymptotic normalisation coefficient (ANC) and, as such, is an indirect determination of the astrophysical cross-section.

We note that Ouellet et al.[17] state in comment b of their Table II that $\phi_{12}$ "could not be determined from data" of the angular distributions measured at 2.2 and 2.0 MeV. Plag et al.[21] and Makii et al.[20] measured $\phi_{12}$ values that agree with equation (1) at energies below 1.5 MeV. Unfortunately, they did not measure over the $1^-$ resonance, where agreement with equation (1) should be used to test the precision of measured angular distributions, as noted by Brune[28] and discussed by us[31].

We underline the effort of the Stuttgart group that measured with a large beam current of $\sim 450 \, \mu A$[18] over many weeks of beam on target that was depleted of $^{13}C$ by a factor of 100, and used the $4\pi$ EUROGAM array of HPGe detectors[19]. The disagreement of these data with the predicted $\phi_{12}$ in spite of the effort to optimise all experimental conditions leads us to conclude that alternative methods should be considered that do not rely on gamma-ray measurements.

Our initial previously published feasibility test, measured with low statistics[31] demonstrated the value of developing an alternative experimental method for studying this important reaction. There, we measured the $^{16}O$ photo-dissociation reaction, the time reverse of the $^{12}C(\alpha, \gamma)^{16}O$ reaction, using a $CO_2$ gas target. The carbon contained in the target introduced a significant background to our measurement, which was removed with software cuts in the data analysis. Like many of the previous works discussed above, our work in ref.[31] suffered similar limitations: the measured $E1$-$E2$ mixing phase angle ($\phi_{12}$) was largely measured with too few statistics across the $1^-$ region to show clear agreement with the prediction from quantum scattering theory, indicating the picture was still not complete.

In this paper, we clearly move beyond the previous results and present new measurements of the photo-dissociation of $^{16}O$ using an $N_2O$ gas target, which physically eliminates backgrounds arising from $^{12}C$. The outgoing charged particles from the reaction were measured using an optical time projection chamber (TPC) detector, with a $2°$ angular resolution, allowing precise angular distributions to be reconstructed. Indeed, the reconstructed angular distributions yield $\phi_{12}$ values that agree with the predictions of quantum scattering theory (equation 1), and resolve the historical disagreement. A further major technical improvement over previous work is the extraction of the centre-of-mass energy of each event individually. This improvement allowed for the cross-section to be evaluated in fine energy steps, circumventing the broad energy spread of the HIγS gamma beam, which had previously been considered as a limitation of this experimental method.

At this point, we dwell only on the quality of our data measured at laboratory energies, and we do not extrapolate to stellar energies. However, our work gives impetus to conduct further measurements at low energies, and such measurements are currently in progress using the Warsaw TPC at the HIγS facility of Duke University[35].

## Results
### Experimental procedure
High-intensity circularly polarised photon beams were generated at the HIγS facility[36] using Compton back-scattering, where free electron laser (FEL) light is scattered from relativistic electron bunches. Electrons were accelerated to 1.2 GeV using a linear accelerator and booster ring before injection into a 31.9 m circumference storage ring. As the electrons circulate, they are bunched by radiofrequency cavities and pass through optical-klystron-5 (OK-5) helical wigglers, which produce circularly polarised light. The photons generated in the wigglers propagate along an optical cavity, sealed by FEL mirrors. Electron bunches interact with these photons, resulting in micro-bunching and coherent FEL light production. This light is Compton back-scattered by subsequent electron bunches, producing high-energy photon beams that are collimated by lead into a beam of 10.5 mm diameter before entering the TPC detector.

Accurate monitoring of the beam intensity was achieved through two complementary methods. The first[37] utilised an in-beam 5-paddle plastic scintillator system to record relative flux. To obtain the absolute flux, a heavily attenuated beam (using six copper absorbers totalling up to 39.35 cm thickness) was also measured using a high-efficiency 10-inch NaI(Tl) scintillator detector. Since the relative flux was simultaneously monitored using the plastic scintillators, this established a paddle/NaI ratio of $\sim 70$. In other words, $\sim 1$ in 70 photons were detected in the paddles and thus removed from the beam. This allowed the normalisation of the paddle measurements to the total photon flux. These results were verified by using a second method: extrapolating the attenuated NaI(Tl) detector counts to full flux using the known attenuation coefficients of the copper absorbers. The beam intensity was typically $10^8 \gamma \, s^{-1}$.

The energy profiles of the photon beams were measured directly by impinging the attenuated beams onto a large (120%) high-purity germanium (HPGe) detector. The detector's spectral response was then unfolded

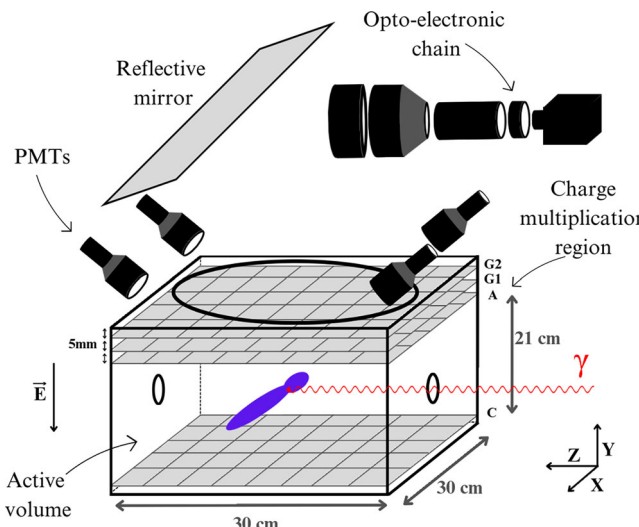

**Fig. 1 | Illustration showing an overview of the OTPC.** The (red) $\gamma$ beam enters the active volume, causing photo-dissociation into (purple) particle tracks. Charge multiplication avalanche grids are shown at the top of the active volume, along with the (98%) reflective mirror, and the readout system (PMTs and the opto-electronic chain).

based on Geant4 Monte-Carlo simulations to obtain the true energy profile of the incident photons[38]. The measured beam energy distributions were best modelled by skewed Gaussian distributions with means of 9.38, 9.49, 9.68, and 9.83 MeV, and an average resolution of 365 keV (FWHM). The full parameterisations of the beam energy distributions, beam intensities, and their uncertainties are included in the supplementary data.

The OTPC is an *active target* detector designed to track decay products from nuclear photo-dissociation reactions with high directional resolution. The construction and operation are discussed in depth in earlier work[39] and a schematic diagram is shown in Fig. 1. It comprises a $30 \times 30 \times 21$ cm$^3$ gas-filled drift volume, operated at 100 Torr with an 80% $N_2O$ plus 20% $N_2$ mixture. Since $N_2O$ is an electron-attaching gas, a relatively low 1000 V drift voltage was applied at the cathode grid to mitigate the effects of electron-$N_2O$ resonances that result in a reduction in the collection efficiency, which would lead to poor energy resolution.

High-energy photon beams enter the active volume through a 15-mm-radius Kapton window. The resulting $^4$He and $^{12}$C ions from the $^{16}$O photo-dissociation ionise gas molecules along their paths, creating electron clouds. These drift upwards under a 50 V/cm uniform electric field, maintained by a drift cage consisting of 66 copper strips (2.5 mm wide, 0.4 mm spacing) embedded in a PCB enclosing the active volume. An avalanche grid system then multiplies the electrons before collection, producing $N_2$ scintillation light. The readout system consists of an anode grid recording total collected charge, photomultiplier tubes (PMTs) for collecting scintillation light for vertical direction information, and mirrors, deflecting (98% reflectivity) light onto an opto-electronic chain to focus an image of the track onto a CCD camera. The combination of the in-plane recording of the track by the CCD camera and the out-of-plane measurements by the PMTs of the time projection enables three-dimensional track reconstruction with scattering angles determined with 2° accuracy.

### Background rejection and track reconstruction

Background events recorded in the OTPC include Cosmic rays, delta electrons, $^{14}$N$(\gamma, p)$, $^{16}$O$(\gamma, n)$, and $^{17/18}$O$(\gamma, \alpha)$. Cosmic rays and $(\gamma, n)$ events were easily removed by an energy threshold placed on the grid (total charge) signal. To eliminate $(\gamma, p)$ events, gates were applied to the total reconstructed track length, integrated CCD pixels, and grid signal. The remaining $^{16}$O$(\gamma, \alpha)$ and $^{17/18}$O$(\gamma, \alpha)$ were separated using a track length cut. To facilitate this, a cut was first placed on the event vertex position along the beam axis to ensure that the beam-induced tracks being considered were fully contained

in the TPC. This procedure ensured that there was no overlap in the track length spectrum for each category of event. Additional gates were applied to all remaining events, considering the absolute distance of a track to the beam axis. The distribution of track vertices allowed the width and direction of the beam to be determined. Tracks with both of their end points residing more than 20.4 mm from the central beam axis were not considered as beam-induced reactions, and were removed from further analysis. A more detailed discussion of this procedure, along with figures, may be found in the methods section.

Unlike in our earlier work[31], the complicated lineshape analysis to remove $^{12}$C photo-dissociation events was not required in the present analysis, greatly reducing computation time and eliminating a key background channel. Instead, theoretical lineshapes for the reaction products were used solely to extract the out-of-plane angle, $\beta$, of the track. This was achieved by fitting the Bragg curves of an $\alpha$ particle and $^{12}$C ion to the in-plane image of the track. The procedure is highlighted in more detail in the methods section. Events with $\beta > 42°$ were removed from analysis. These tracks have too short a projection in-plane, which made it difficult to reliably extract the in-plane angle, $\alpha$. Events with $\beta < 20°$ were also excluded as the ability to accurately extract the $\beta$ angle diminishes as $\beta$ approaches 0°. The in-plane angle $\alpha$, obtained from a RANSAC (Random Sample Consensus) fit to the CCD image, was combined with the out-of-plane angle $\beta$ to obtain the scattering angle $\theta$. This was then boosted to the centre-of-mass (CM) frame using the known momentum of the incident photon beam, to produce angular distributions of $\theta_{cm}$. Monte-Carlo simulations were used to calculate the detection efficiency as a function of angle $\theta$, to correct the angular distributions for fiducial cuts and resolution effects.

### Integrated cross-section analysis

For cross-section measurements, we used the total $^{16}$O dissociation event counts recorded in the TPC before angular cuts. The track length for each event was used, in conjunction with stopping power tables, to calculate the reaction centre-of-mass energy. The recorded centre-of-mass energy spectrum in the TPC was used along with the measured beam energy distribution to split the total cross-section across the width of the beam. Rather than obtaining an effective total cross-section due to excitation across the width of the whole $\gamma$ beam, as was done in previous work[31], this advancement permits the cross-section to be measured in finer energy steps. The measured photo-dissociation cross-sections were converted to the forward capture reaction cross-sections using a detailed balance factor. The deconvolution method of Brune and Sayre[40] was then employed to correct for the TPC energy resolution of 162 keV FWHM. Note the improved resolution versus the 365 keV beam energy spread. This full process and the relevant error propagation are explained in greater detail in the methods section. Figure 2 compares the measured total cross-sections with the world data. The data are given in Table 1.

### Angular distribution analysis

To be useful to the astrophysics community, the measured angular distributions along with the measured energy profiles should be incorporated into a global fit, such that the underlying $E1$ and $E2$ cross-sections can be extracted. However, for the purposes of this paper, the centre-of-mass angular distributions were fitted with a modified version of the $E1-E2$ multipole mixing formula, originally given by Dyer and Barnes[15], in order to extract energy-averaged parameters that can be compared with theoretical predictions. The original formula is defined as

$$
\begin{aligned}
W(\theta) = {} & (3|A_{E1}|^2 + 5|A_{E2}|^2)P_0(\cos\theta) \\
& + \left(\frac{25}{7}|A_{E2}|^2 - 3|A_{E1}|^2\right)P_2(\cos\theta) \\
& - \frac{60}{7}|A_{E2}|^2 P_4(\cos\theta) \\
& + 6\sqrt{3}|A_{E1}||A_{E2}|\cos\phi_{12}\left[P_1(\cos\theta) - P_3(\cos\theta)\right],
\end{aligned}
\tag{2}
$$

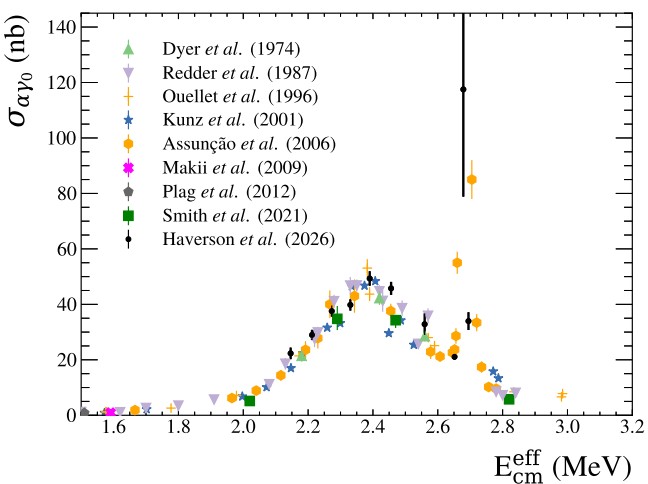

**Fig. 2 | Total cross-sections of the $^{12}$C($\alpha,\gamma_0$) reaction.** The cross-sections measured in this work (black circular points) are compared with previous world data (different coloured/shaped points). Details of the analysis and error bars are discussed in the Integrated cross-section analysis part of the paper and the Total cross-section part of the Methods section.

where $A_{E1}$ and $A_{E2}$ are the E1 and E2 amplitudes, and $\phi_{12}$ is the mixing phase angle. This equation was modified so as to be parameterised as a ratio of cross-sections,

$$W(\theta) = W_{E1}(\theta) + \frac{\sigma_{E2}}{\sigma_{E1}} W_{E2}(\theta) + \sqrt{\frac{\sigma_{E2}}{\sigma_{E1}}}\cos(\phi_{12})W_{12}(\theta). \quad (3)$$

The terms in equation (3) are defined as

$$W_{E1}(\theta) = P_0(\cos(\theta)) - P_2(\cos(\theta))$$

$$W_{E2}(\theta) = P_0(\cos(\theta)) + \frac{5}{7}P_2(\cos(\theta)) - \frac{12}{7}P_4(\cos(\theta))$$

$$W_{12}(\theta) = \frac{6}{\sqrt{5}}\left(P_1(\cos(\theta)) - P_3(\cos(\theta))\right),$$

where $P_n$ are the Legendre polynomials.

For all but the lowest beam energy considered in this study, the very narrow ($\Gamma = 0.625$ keV) $2^+$ resonance at 9.845 MeV in $^{16}$O was populated due to the broad gamma beam. Therefore, the measured angular distributions are a combination of equation (3), which represents the underlying behaviour, in addition to a pure E2 contribution from the narrow $2^+$ level. Therefore, to permit a meaningful angular distribution analysis, depending on the proximity of the beam energy to this narrow $2^+$ resonance, a proportion of pure E2 angular distribution was added incoherently to equation (3), so as not to perturb the extracted energy dependence of $\sigma_{E2}/\sigma_{E1}$ and $\phi_{12}$. The Methods section contains further details on how the amount of E2 was determined and its uncertainty.

When fitting the extracted angular distributions, the fiducial efficiency correction was incorporated into the fit function to preserve Poisson statistics. The fit function was also multiplied by a $\sin\theta$ term before comparing with the angular data to account for solid angle effects. A 1°-binned negative log-likelihood minimisation was employed, using the *Minuit* minimiser, varying $\phi_{12}$ and $\sigma_{E2}/\sigma_{E1}$. Figure 3 presents the fitted angular distributions, with the minimised fit function scaled and overlaid on the binned efficiency-corrected data for visualisation purposes.

Table 2 presents the best-fit parameters for each effective CM energy, including statistical uncertainties and goodness-of-fit metrics. The extracted $\phi_{12}$ values are shown in Fig. 4 and compared with energy-averaged predicted values of $\phi_{12}$ from elastic scattering data. Clear agreement is observed between the measured $\phi_{12}$ values with the predicted values, unlike the case for many of the current world data.

### Table 1 | List of extracted total cross-sections

| $E_{cm}$(MeV) | $\sigma_{photo}$ (nb.) | $f_{db}$ | $\sigma_{capture}$ (nb.) |
|---|---|---|---|
| 2.145 ± 0.019 | 1544 ± 145 | 69.16 ± 0.11 | 22.3 ± 2.1 |
| 2.211 ± 0.012 | 2036 ± 131 | 70.29 ± 0.06 | 29.0 ± 1.9 |
| 2.272 ± 0.011 | 2679 ± 143 | 71.30 ± 0.04 | 37.6 ± 2.0 |
| 2.330 ± 0.009 | 2874 ± 148 | 72.22 ± 0.01 | 39.8 ± 2.1 |
| 2.390 ± 0.008 | 3609 ± 192 | 73.17 ± 0.01 | 49.3 ± 2.6 |
| 2.456 ± 0.007 | 3392 ± 174 | 74.14 ± 0.02 | 45.8 ± 2.4 |
| 2.558 ± 0.008 | 2484 ± 295 | 75.62 ± 0.11 | 32.8 ± 3.9 |
| 2.652 ± 0.005 | 1622 ± 70 | 76.90 ± 0.01 | 21.1 ± 0.9 |
| 2.679 ± 0.004 | 9079 ± 2990 | 77.26 ± 0.01 | 117.5 ± 38.7 |
| 2.695 ± 0.006 | 2631 ± 249 | 77.46 ± 0.01 | 34.0 ± 3.2 |

The $\sigma_{photo}$ are deconvolved photo-dissociation cross-sections, $f_{db}$ are the detailed balance conversion values, and $\sigma_{capture}$ are the converted capture cross-sections.

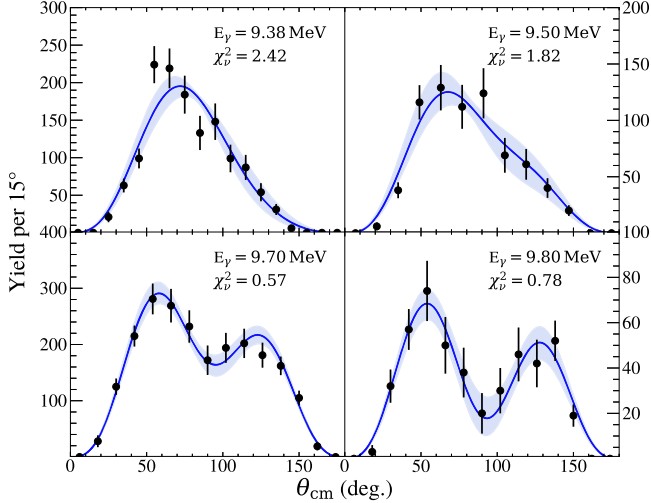

**Fig. 3 | Measured angular distributions of the $^{12}$C($\alpha,\gamma_0$)$^{16}$O reaction.** Experimental data (black points) are plotted in 15° bins, include efficiency corrections, and are presented at the shown effective centre-of-mass energies. Error bars are efficiency-corrected $1\sigma$ SD statistical errors. The lines show the two-parameter fit $\left(\frac{\sigma_{E2}}{\sigma_{E1}},\phi_{12}\right)$ of the partial wave decomposition, along with the shaded $1\sigma$ SD error band of the fit.

### Table 2 | Summary of energy-averaged fit parameters

| $E_\gamma^{nom}$ (MeV) | $E_{cm}^{eff}$ (MeV) | $\phi_{12}$ (deg.) | $\frac{\sigma_{E2}}{\sigma_{E1}}$ | $\chi_\nu^2$ |
|---|---|---|---|---|
| 9.38 | 2.308 (8) | $39.96^{+11.97}_{-38.25}$ | $0.11^{+0.09}_{-0.06}$ | 2.42 |
| 9.50 | 2.394 (10) | $58.34^{+10.87}_{-29.99}$ | $0.15^{+0.16}_{-0.12}$ | 1.82 |
| 9.70 | 2.570 (7) | $72.22^{+7.39}_{-8.94}$ | $0.25^{+0.16}_{-0.13}$ | 0.57 |
| 9.80 | 2.660 (3) | $78.68^{+8.26}_{-8.85}$ | $2.93^{+2.44}_{-1.18}$ | 0.78 |

The parameters were extracted using the binned maximum log likelihood fits to the angular distributions. Correlations between the fit parameters are accounted for in the quoted uncertainties.

The quoted uncertainties include the dominant statistical errors from the fits, extracted from contour plots that track log likelihood as a function of both $\sigma_2/\sigma_1$ and $\phi_{12}$ to account for correlations. They also include the uncertainty introduced by extracting the additional E2 contribution from the narrow $2^+$ state, which is accounted for in the fits. The relative contribution from each source of error varies depending on the beam energy. A full error quota per energy is included in the supplementary data.

## Conclusions

The cross-section of the $^{12}$C($\alpha$, $\gamma$) reaction has been recognised as perhaps the single most important nuclear input to stellar evolution theory. However, its value at astrophysical energies, in the Gamow window at 300 keV, relies on extrapolating cross-sections at higher energies, which have eluded accurate direct measurements over the last five decades. Discrepancy of alpha capture data on the $E1-E2$ mixing phase angle ($\phi_{12}$) with an elementary prediction of quantum mechanics leads us to conclude significant systematic uncertainties in existing data. Fixing $\phi_{12}$ to its predicted values in the analyses of historical data changes the ratio of the $E2/E1$ partial cross-sections by up to a factor of 4. Thus, these data should not be included in extrapolations to stellar energies.

Here we present compelling evidence for accurate measurements of the cross-section of the $^{12}$C($\alpha$, $\gamma$) reaction, employing a TPC detector operating in gamma beams. Most importantly, we show, using our alternative method, clear confirmation of the elementary prediction of the $E1-E2$ mixing phase angle and demonstrate our ability to measure angular distributions at high energies with high accuracy. Furthermore, we present a significant step forward in the analysis by using the TPC to measure the centre-of-mass energy of each event individually. This completely overcomes limitations pertaining to the finite energy width of the $\gamma$ beam, permits the energy dependence of the cross-sections to be measured with higher precision, and

opens up the possibility to conduct such measurements at other broad-energy $\gamma$ beam facilities. These initial measurements serve as a strong impetus to continue these challenging measurements to lower energies to allow for accurate extrapolations of the $E1$ and $E2$ cross-sections to the relevant stellar conditions, without relying on indirect methods.

## Methods

The OTPC enables three-dimensional reconstruction of the $^{16}$O($\gamma$, $\alpha$)$^{12}$C reaction by measuring the total energy deposited, momentum, and angular distributions of the emitted particles, which also allows for particle identification. Thus, the OTPC facilitates the separation of different reaction channels using three main tools: the CCD camera photograph, the total energy deposition for each event on the grids, and the time projection of each particle track detected with the PMTs. Further details on the analysis methods are provided below.

### Image analysis

1The OTPC uses a CCD camera to capture 8-bit grey-scale images of the $x$-$z$ horizontal plane, triggered by a leading edge discriminator set at 800 keV for charge collected on the anode grid. The process of cleaning these images during the data analysis is depicted in Supplementary Fig. 2. Each run commenced with a blank exposure of the CCD camera, triggered externally without the gamma beam, to obtain a reference image for background subtraction. This "flat-field" correction is applied to all subsequent events in the run by subtracting this blank exposure. Supplementary Fig. 2A illustrates an example of a flat-field-corrected $^{16}$O($\gamma$, $\alpha$) image.

To optimise processing time, the image is then pixelated horizontally and vertically by a factor of 4, as shown in Supplementary Fig. 2B. Using this compressed image, the mean background pixel value and standard deviation are calculated. A threshold is set at 6 standard deviations above the mean background, resulting in the image shown in Supplementary Fig. 2C. To isolate the main particle track, a recursive clustering algorithm was implemented. This scans the compressed image in a $3 \times 3$ grid, zeroing pixels with fewer than 5 non-zero neighbouring pixels. The resulting cleaned image (Supplementary Fig. 2D) serves as a mask, applied to the flat-field corrected image to restore the original resolution (Supplementary Fig. 2E). Finally, all charge outside of the largest cluster is zeroed, and a final finer neighbour scan is applied to the restored image, thus fully cleaning the track (Supplementary Fig. 2F).

### Angle reconstruction

The fully cleaned images (Supplementary Fig. 2F) were fit with a line using a RANSAC algorithm, as shown in Fig. 5a. The RANSAC method was selected as it is designed to reduce the impact outliers have on the minimisation. This process defines the in-plane angle $\alpha$. The image was then projected onto this fit line, as shown in Fig. 5b, and the resulting projection

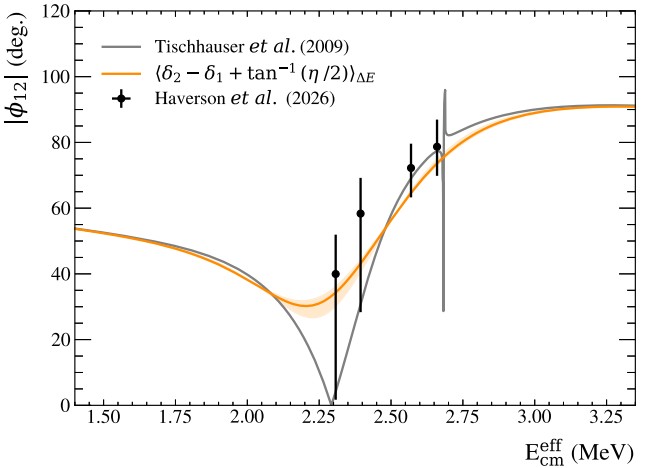

**Fig. 4 | Results of the angular distribution analysis.** The $E1-E2$ mixing phase angles ($\phi_{12}$) measured in this analysis (black points) compared with the energy-averaged theoretical prediction (solid orange line with shaded $1\sigma$ SD uncertainty band). The error bars are dominated by $1\sigma$ SD statistical errors on the angular distribution fit with small contributions from the uncertainty on the additional $E2$ factor (added to account for the narrow $2^+$ resonance at 9.845 MeV in $^{16}$O).

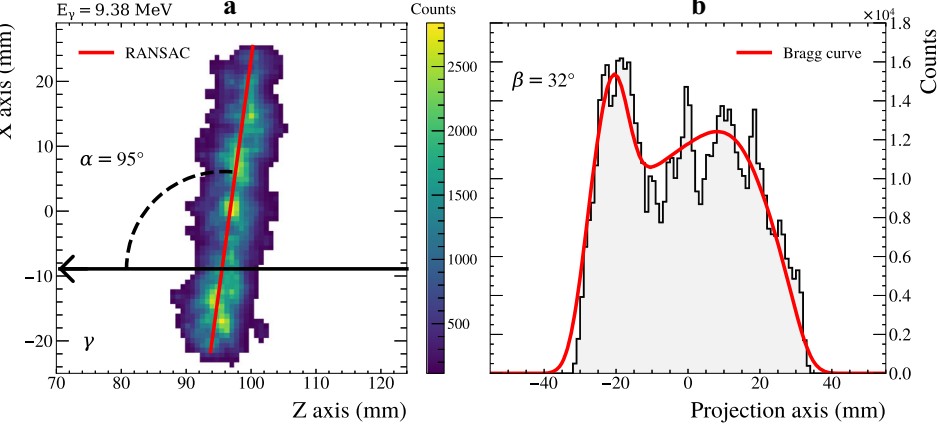

**Fig. 5 | Example reconstructed $^{16}$O event. a** An example fully cleaned $^{16}$O event. The red line is the RANSAC fitted line, the gamma beam direction is indicated by the black arrow and the $\alpha$ angle is shown. **b** The projection of the event onto the RANSAC line, and a Bragg curve fit used to extract the out-of-plane $\beta$ angle.

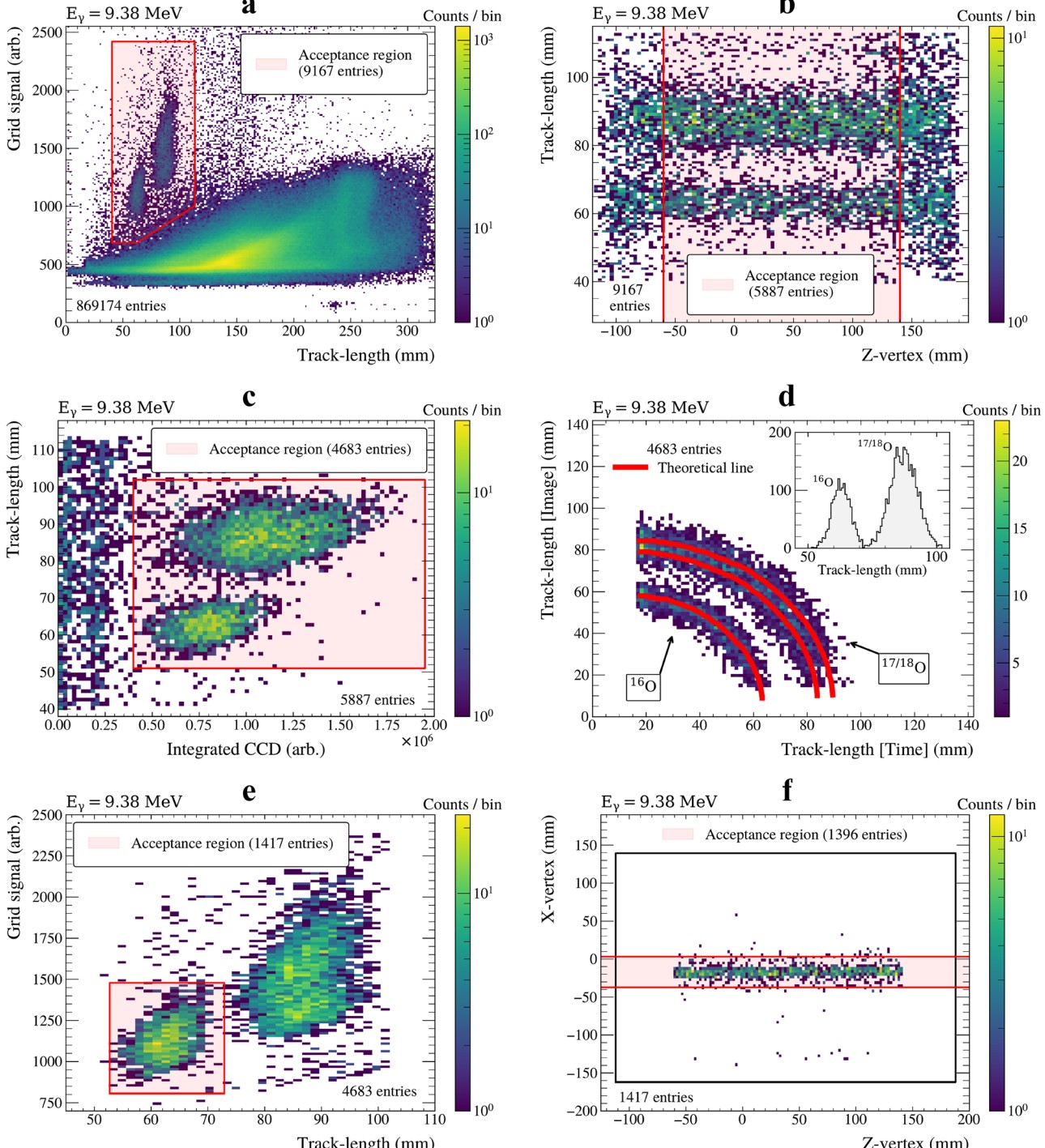

**Fig. 6 | Data reduction and verification steps.** Each histogram shows a step in analysing $E_\gamma$ = 9.38 MeV. The total number of events before and after each cut is given, with the acceptance region for cuts highlighted in red. **a** Track length vs. deposited energy: an initial wide acceptance region is placed around all oxygen events. **b** z-vertex position vs. measured total track length: distribution of event vertices along the beam axis within the TPC. Events that are not fully contained are removed. **c** Cluster-cleaned CCD image pixel count vs. total track length: the $^{16}O$ and $^{17/18}O$ are separated from the $(p, \gamma)$ nitrogen background. **d** Track length in time vs. track length in image: Verification of decay channel assignment through tracking of track length for events at different angles in the TPC. Red lines overlayed are calculations of expected track lengths. The inset shows the total track length histogram, demonstrating clear isolation of $^{16}O$ dissociation events. **e** Track length vs. deposited energy: another acceptance region placed over $^{16}O$ candidate events. **f** z-vertex positions vs. x-vertex positions: The distribution of event vertices within the TPC, where the black rectangle defines the active area. A Gaussian fit to the x-vertex distribution defines the spatial beam profile in the TPC. Any event with an endpoint more than $3\sigma$ away from the mean vertex position was subsequently removed from the analysis.

was fit with a series of Bragg curves, each corresponding to a different out-of-plane $\beta$ angle. The larger the angle $\beta$, the shorter the Bragg peaks will appear. The theoretical lineshapes were generated in 1° $\beta$ steps and convolved with a Gaussian, which models the CCD resolution. The lineshape giving the best

fit provided the out-of-plane angle $\beta$. The extracted $\alpha$ and $\beta$ angles were combined to obtain the polar angle $\theta$ in the laboratory frame, which was then boosted into the centre-of-mass frame using the momentum of the gamma beam.

## Reaction channel selection

Data selection involved several cuts to isolate events of interest. Figure 6a displays all data collected for the nominal beam energy $E_\gamma = 9.38$ MeV, showing reconstructed total track length plotted against the grid signal (proportional to energy). The track length was calculated by combining the length in the horizontal plane, extracted at image processing stage E, with the vertical length extracted from the time projection. This plot reveals a large region of events corresponding to photo-dissociation of nitrogen, along with two distinct clusters corresponding to the expected track lengths of $^{16}O$ and $^{17/18}O$ dissociation in 100 Torr $N_2O+N_2$. To isolate these $(\gamma, \alpha)$ events, a graphical cut was first applied to Fig. 6a in order to remove most of the nitrogen background.

Once the $(\gamma, \alpha)$ events were isolated, a cut on the vertex position of each event was placed along the $z$-axis. Although $^{17/18}O$ tracks are longer, if part of the track escapes the detector, it will appear shorter, and hence may be wrongly classified as an $^{16}O$ event. A vertex cut along the $z$-axis (beam axis) ensured the events were fully contained within our detector and thus removed this possible overlap in measured track lengths, which is shown in Fig. 6b.

Further background was removed through analysis of the integrated cluster-cleaned CCD images (Supplementary Fig. 2F). The fragmented proton tracks from $(\gamma, p)$ events, caused by to their lower stopping power combined with the low gas gain, undergo more severe "cleaning" than $(\gamma, \alpha)$ events. This results in an image with a much smaller remaining cluster, which, when integrated, leads to lower charge, producing clear separation in the integrated CCD charge distribution, as shown in Fig. 6c. An additional track-length selection was implemented at this stage. The whole cut is shown by the shaded region in Fig. 6c.

The remaining oxygen event candidates were validated by decomposing the track length. Figure 6d shows the reconstructed track length decomposed into both the time and image projections (vertical and horizontal). The red lines represent theoretical calculations corresponding to each of the reaction channels, which include resolution effects, verifying our pixel-to-mm and time-to-mm calibrations. The $^{16}O$ events were then isolated from $^{17/18}O$ data by placing $3\sigma$ cuts to total track length and grid signal spectra, as shown in Fig. 6e.

The final cut set a limit on the absolute distance that the event originates from the beam axis, defined as the $x$-vertex position. The distribution of vertex positions in the horizontal $x$-$z$ plane for the 9.38 MeV data is shown in Fig. 6f. These data were projected onto the $x$-axis and fitted with a Gaussian ($\sigma = 6.8$ mm). Events residing more than $3\sigma$ from the centre of this distribution were removed from further analysis as they were not considered to have originated from the beam.

## Interpreting angular distributions

As noted earlier in the paper, due to the broad gamma beam, the very narrow $2^+$ resonance at 9.85 MeV in $^{16}O$ was generally populated for most incident beam energies. This poses a challenge for interpreting the measured angular distributions, since the shape of the angular distribution will not only be dictated by the underlying $\sigma_{E2}/\sigma_{E1}$ partial cross-sections at that energy, but will also include some amount of pure $E2$ shape, corresponding to when the narrow $2^+$ state is populated. Therefore, the angular distributions were fit using a function comprising a combination of equation (3), which represents the underlying behaviour, with the incoherent addition of a pure $E2$ contribution, as $W(\theta) = (a-1)W(\theta) + aW_{E2}(\theta)$, where $a$ is the proportion of pure $E2$.

To determine the proportion of events that corresponded to the population of the narrow $2^+$ state, centre-of-mass energy spectra were examined. Track lengths were converted to particle energies using SRIM energy loss calculations. Then, using the photo-dissociation kinematics and the measured scattering angle, centre-of-mass energies were calculated. This process offered better energy resolution ($\sigma = 69$ keV) than using the grid energy signal, since track lengths were measured with higher precision. The centre-of-mass energy spectra are shown in Fig. 7. These were fitted with

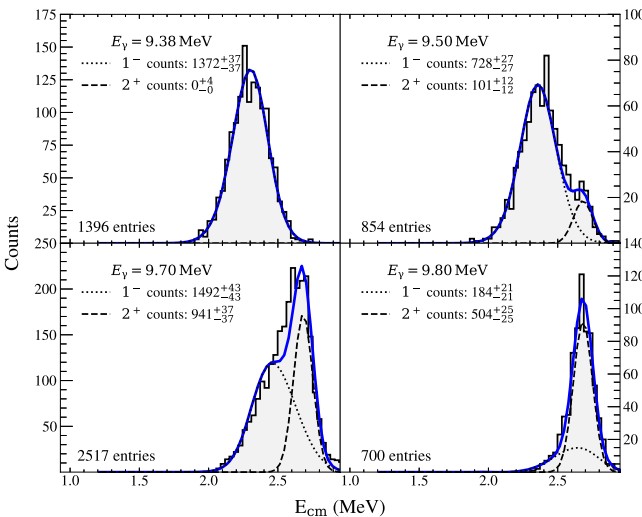

**Fig. 7 | Histograms of centre-of-mass energy for $^{16}O$ events at each beam energy.** Fits are obtained by modelling the narrow $2^+$ state and background, dominated by the broad $1^-$ state, as described in the text.

two peaks: one to model the underlying cross-section, and the other corresponding to the narrow $2^+$ state.

To ascertain the shapes of these peaks, and constrain the fits, lineshapes for the narrow $2^+$ and broader underlying cross-section feature (predominantly the $1^-$ state) were generated using the AZURE2 $R$-matrix code. The shape of the total cross-section was phenomenologically modelled as a simple sum of the two resonances, as shown in Supplementary Fig. 3. These two lineshapes were each folded with the measured energy profiles of the $\gamma$ beam, convolved with the TPC detector resolution, and normalised. They were then fit to the centre-of-mass energy spectra, and the relative peak heights were used to extract the proportion of events corresponding to the $2^+$ state. The proportions of these two peaks and their uncertainties are provided in the supplementary data.

The highest energy angular distribution was fit with the standard angular distribution formula, equation (3), without the added $E2$ component. At this beam energy, the narrow $2^+$ state is expected to be very strongly populated. At this highest energy, the $\sigma_{E2}/\sigma_{E1}$ value from the fit to the angular distribution agrees with the proportion of pure $E2$ extracted from the fit to the centre-of-mass energy spectrum. From the angular distribution fit $\sigma_{E2}/\sigma_{E1} = 2.93^{+2.44}_{-1.18}$. From the fit to the total energy spectrum $\sigma_{E2}/\sigma_{E1} = 2.74 \pm 0.34$. This agreement demonstrates that the addition of an incoherent $E2$ is an appropriate analysis method for the other energies.

## Energy averaging parameters

When comparing the extracted $\phi_{12}$ values with theoretically predicted values, the energy spread of the beam must be accounted for. Since the extracted $\phi_{12}$ is located in the cross term of equation (3), its energy-averaged value is affected by the energy dependence of $\frac{\sigma_{E2}(E)}{\sigma_{E1}(E)}$. The energy-averaged value may be calculated as ref. 40

$$\langle \cos\phi_{12} \rangle = \frac{1}{\sqrt{\langle\sigma_{E2}\rangle\langle\sigma_{E1}\rangle}} \qquad (4)$$
$$\times \int \cos\phi_{12}(E)\sqrt{\sigma_{E2}(E)\sigma_{E1}(E)}G(E)dE,$$

where $G(E)$ is the normalised beam energy profile. The angular distribution fitting procedure aims to examine the underlying $E1-E2$ interference by accounting for an incoherent addition of a pure $E2$ contribution. Assuming that the $\frac{\sigma_{E2}(E)}{\sigma_{E1}(E)}$ ratio of the underlying behaviour does not change significantly across the beam width, the $\sqrt{\sigma_{E2}(E)\sigma_{E1}(E)}$ term may be assumed to be

constant and taken out of the integral. The formula, therefore, becomes

$$\langle \cos \phi_{12} \rangle \approx \int \cos \phi_{12}(E) G(E) dE. \tag{5}$$

Using the above formula, the $\phi_{12}(E)$, as given by the elastic scattering data[24], was energy-averaged across the gamma beam, allowing a comparison with the experimentally measured values from the present study. The comparison is shown in the main text in Fig. 4. The error band on the energy-averaged $\phi_{12}$ line accounts for the uncertainties in the shape of the beam energy profiles.

## Total cross-section

The measured centre-of-mass energy spectra in Fig. 7 were binned more coarsely to improve statistical uncertainties in each bin. Thalue may be calcuese were then used to calculate the total cross-section for each centre-of-mass energy bin. A relative cross-section may be calculated on a bin-by-bin basis by simply dividing the measured energy spectrum by the corresponding beam energy profile. The absolute photo-dissociation cross-section (energy averaged across each bin) is calculated as

$$\langle \sigma_{\gamma\alpha_0}(E_i) \rangle = \frac{S(E_i)/G(E_i)}{\mathcal{L} \, \epsilon \, f_{LT}}, \tag{6}$$

where $S(E_i)$ is the binned centre-of-mass energy spectrum as measured in the TPC, $G(E_i)$ is the normalised binned beam energy distribution, $\mathcal{L}$ is the beam luminosity, $f_{LT}$ is the TPC live time fraction, and $\epsilon$ is the efficiency accounting for data reduction cuts. The uncertainty on the photo-dissociation cross-sections, $\langle \sigma_{\gamma\alpha_0}(E_i) \rangle$, is a combination of the errors on: the counts in each bin ($\sqrt{N}$), the beam energy profile, and the beam intensity. Further details are contained in the supplementary data.

To compare with world data, the deconvolution method, outlined by Brune and Sayre[40], and used in our earlier work[31], was then employed to account for the resolution of the TPC. The correction factor is a ratio of the theoretical energy-averaged cross-section, $\langle \sigma \rangle_{theo}$, and the theoretical cross-section at the effective energy, $\sigma(E_{eff})$. The effective energy is defined as

$$E_{eff} = \frac{\int E \, H(E) \, \sigma(E) \, dE}{\int H(E) \, \sigma(E) \, dE}, \tag{7}$$

and the energy-averaged cross-section is defined as

$$\langle \sigma \rangle_{theo.} = \frac{\int \sigma(E) \, H(E) \, dE}{\int H(E) \, dE}, \tag{8}$$

where $H(E)$ is an approximation of the detector response, obtained by splitting the probability distribution that models the beam energy profile, as measured by the HPGe, $G(E)$, within the considered energy bin, then convolving with the centre-of-mass energy resolution. The correction factor is therefore given by

$$f_{cor.} = \frac{\sigma(E_{Eff})}{\langle \sigma \rangle_{theo.}}. \tag{9}$$

The error in the correction factor was typically 1% at the lowest energies, and 30% near the narrow $2^+$ resonance, where the cross-section changes rapidly. The uncertainties were dominated by errors in the shape of the beam energy profile. These photo-dissociation cross-sections were converted to the time-reverse capture reaction using the principle of detailed balance, with the detailed balance factor evaluated at the effective energy of each bin. The errors on the centre-of-mass energies, quoted in Table I, which are due to uncertainties in the energy calibration, lead to a small uncertainty in the detailed balance factor of <1%. The resulting overall uncertainties on the capture cross-sections are given in Table I.

## Rights retention statement

For the purpose of open access, the author has applied a Creative Commons Attribution (CC BY) license to any Author Accepted Manuscript version of this paper arising from this submission.

## Data availability

High-level data pertaining to total cross-sections, angular distributions, angular efficiency corrections, angular distribution fitting, beam intensities and parameterisation of beam profiles are provided in the supplementary data. For all $\gamma$-beam energies, lists of the measured $\theta$ angles and centre-of-mass energies are provided to permit an unbinned maximum likelihood analysis of the angular distributions and re-evaluation of the cross-sections. The raw data have also been made publicly available for re-analysis under a Creative Commons BY-NC 4.0 license. The image files of tracks on an event-by-event basis, the ROOT files containing the time projections and energy signals, and the ROOT files containing the beam intensity monitoring data are provided. All data may be found on the Sheffield Hallam University data repository at: https://doi.org/10.17032/shu-0000000309.

## Code availability

Example analysis codes, written in ROOT C++, are available in this data repository: https://doi.org/10.17032/shu-0000000309.

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

## Acknowledgements

This work was supported by the UK Science and Technology Facilities Council (STFC) [grant number ST/Y000331/1]; and the U.S. Department of Energy, Office of Science, Office of Nuclear Physics [grant numbers DE-FG02-94ER40870, DE-SC0005367, DE-FG02-97ER41033 and DE-FG02-91ER-40608]. The authors acknowledge the efforts of the UConn-TUNL collaboration for the collection and dissemination of data. We thank the staff of HI$\gamma$S at Triangle Universities Nuclear Laboratory for the operation of the facility, and C.R. Howell and M.W. Ahmed for in-depth discussions and assistance with various aspects of the work. One of us (MG) would like to acknowledge a discussion with Marco Limongi, who pointed out Hoyle's paper, where the crucial role of the $^{12}$C($a$,$\gamma$) was discussed for the first time.

## Author contributions

M.G. served as the spokesperson of the UConn-TUNL collaboration. The data were analysed by K.C.Z.H. with the assistance of R.S. and M.G. Beam characterisation support was extensively provided by S.F., with additional support from S.R.S. and D.K.S. All authors contributed to the final manuscript.

## Competing interests

The authors declare no competing interests.
