## [Transparent Peer Review file · Communications Physics]

Precision $^{12}\text{C}(\alpha,\gamma)$ Angular Distribution Measurements: First Significant Physics Result with a Time Projection Chamber Operating in Gamma Beams

Corresponding Author: Mr Kristian Haverson

Version 0:

Reviewer comments:

Reviewer #1

(Remarks to the Author)

My report is given in the attachment.

Reviewer #2

(Remarks to the Author)

The manuscript of Haverson et al. under consideration presents a follow-up paper to Smith et al., Nature Communications 12, 5920 (2021). The 2021 paper presented a new method to address very important experimental uncertainties for nuclear astrophysics associated with the $^{12}\text{C}(\alpha,\gamma)^{16}\text{O}$ reaction rate. The method introduced in 2021 employs gamma beams and a time projection chamber to measure the time-inverse $^{16}\text{O}(\gamma,\alpha)^{12}\text{C}$ reaction, providing some unique advantages over existing methods. The 2021 paper presented a plot of the E1-E2 mixing phase angles measured using the new method demonstrating good agreement with theoretical predictions of quantum mechanics.

The manuscript of Haverson et al. presents new measurements of the same reaction over the same energy range using the same method with some improvements. For example, the fill gas of the TPC has been changed to reduce a significant background and a new technique has been introduced to circumvent the intrinsic energy spread of the gamma beam. The manuscript culminates in a plot of the E1-E2 mixing phase angles similar to the one from the 2021 paper, with modest improvement. The authors elaborate about the importance of the agreement between data and theory in this plot as a quality check of $^{12}\text{C}(\alpha,\gamma)^{16}\text{O}$ data and emphasize that previous data sets from other groups haven't demonstrated agreement.

The astrophysical problem being pursued is a very important one. I believe that this is a nice experimental method, which has been demonstrated to be improved somewhat in the present manuscript. However, I do not believe that there is sufficient technical advancement or novel scientific impact in this manuscript beyond the content of 2021 paper (or content that could have been added to the 2021 paper at the time without the need for additional acquisition of data) to recommend publication in Communications Physics.

Reviewer #3

(Remarks to the Author)

The paper "Precision $^{12}\text{C}(\alpha,\gamma)$ Angular Distribution Measurements: First Significant Physics Result with a Time Projection Chamber Operating in Gamma Beams" presents the measurement of the differential cross sections (angular distributions at four energies) of the reaction $^{16}\text{O}(\gamma,\alpha)^{12}\text{C}$ in a time projection chamber using N_2O gas as a target. This was specifically done in the region of a 1- resonance in the $^{12}\text{C}(\alpha,\gamma)$ reaction around $E_{\text{cm}} = 2.4$ MeV, where previous measurements delivered inconclusive results. In addition, total cross sections were extracted over the resonance. This work is a continuation of an existing campaign to develop alternative methods to measure this elusive nuclear reaction; modifications to the setup eliminated previously observed beam induced background.

The work lays out strong evidence for the feasibility of this method and is certainly a good motivation to continue improvements and further measurements with gamma beams and time projection chambers - for example the uncertainty in the two low energy points of the angular distribution should be improved to more strongly constrain the model. As such it is of interest to the broader nuclear astrophysics community (and more) and in principal worthy of publication. The used method and the description of setup and analysis is sound but there are some issues with the introduction, the organisation of the text and the references that need to be addresses before I can suggest acceptance in Nature Communications Physics. Following are my comments.

1) The introduction rests heavily on the recent publication by JWST on the observation of the compact object in the center of SN 1987A. Besides the citation being only to the press release by JWST and not the actual article in Science, which should be corrected, there appears to be some misunderstanding of the statements in said paper: in the present manuscript it is stressed that JWST ruled out a black hole compact object from their new observations, hence giving lead to a necessary revision of the mass cutoff. This is not what is written in the JWST article. Instead, JWST at the beginning of their discussion rules out a black hole simply due to mass considerations and then works only with various neutron star models. Nowhere due they rule out a black hole from the new observations.

This point does not diminish the importance of cross section measurements of $^{12}\text{C}(\alpha, \gamma)$, it is after all still one of the most important nuclear reactions in astrophysics, but it appears somewhat odd that the present text relies so heavily on what appears to be some misunderstanding. This can easily be corrected though.

2) The organisation of the text is a bit confusing, with the discussion section being in the middle of the text, followed by the methods section. Unless this is a requirement by the journal it seems more natural and easier to follow if the methods section accompanies the presentation of the results and the manuscript concludes with the discussion.

3) While the authors very nicely describe the analysis in the methods section, a better discussion of systematic uncertainties would be helpful, also in order to judge possible further improvements and intrinsic limits of the method.

4) The references are a using a mixture of different styles, something First Name, Last Name, sometimes the other way around. Fully spelled out first names are chosen for some references while abbreviated ones for others. The titles are included or are omitted, without discernible reason for either choice. In case the titles are included, the separator between the last author and the title changes from reference to reference or is not present at all. Lastly, in reference [31] the name of an author is clearly spelled wrong.

Version 1:

Reviewer comments:

Reviewer #1

(Remarks to the Author)

The authors have addressed all of my previous comments to my satisfaction. I have no further comments.

Reviewer #3

(Remarks to the Author)

The authors have addressed my concerns and I now can support publication in Nature Communications Physics.

We would like to thank the three reviewers for their very constructive comments and suggestions that have improved our manuscript.

The comments of the reviewers are given in **BLACK** and our responses in **BLUE**.

Due to changes to the manuscript, numbered references have shifted. In this response, references corresponding to the numbering in the earlier draft are accompanied with *.

Reviewer 1

The paper has no conclusion section, which seems to be an odd oversight. Section IV, labeled “Discussion” seems to essentially be this however.

The Nature Communications Physics journal format does not contain a dedicated conclusion section, but instead the conclusions are included in what is labelled as the discussion section. In response to your comment, we have bolstered our discussion section to emphasise the concluding remarks, following the format of Nature Communications Physics.

Are there other experiments that would strengthen the paper further? How much would they improve it, and how difficult are they likely to be?

Yes, experiments with improved precision would greatly strengthen the authors’ arguments. It is unclear if they are feasible at this time.

A continuation of this measurement is indeed feasible. Together with the team at the University of Warsaw, we used the Warsaw electronic TPC to continue this measurement to lower energies with a measurement discussed in [35] – a conference proceedings publication of our collaboration with the Warsaw team that show preliminary angular distributions.

Are the claims appropriately discussed in the context of previous literature?

For the most part, but there are a few places where there are some issues. In particular, the authors cite their own work as motivation, which is not even a published work (reference 28). This seems odd considering there are other works that are published also clearly provide this motivation, for example, the review by Ref. 30.

We understand and appreciate the concern of the reviewer. However, the review in Ref. [28*] is very different than that of [30*]. In deBoer et al. [30*] they refer to angular distributions contained in the Ph.D. thesis of Fey, M. (2004), Im Brennpunkt der Nuklearen Astrophysik: Die Reaktion $^{12}\text{C}(\alpha,\gamma)^{16}\text{O}$, Ph.D. thesis (Universit Stuttgart).

Gai was very troubled by the disagreement of the data of the Stuttgart group with an elementary prediction of quantum mechanics as he discussed in [26*]. This disagreement is so troubling

that we decided to show it in an appendix to this manuscript. Note that all nine low energy data points disagree with the elementary prediction of quantum mechanics. Most obvious is the disagreement on the 1^- resonance where the cross section is large. This disagreement with quantum mechanics was ignored in the intervening 20 years.

To get to the bottom of this disagreement, Gai [28*] examined the very same Fey (2004) Ph.D. thesis, but he examined the raw gamma-ray spectra, not the angular distributions. Fey's thesis presents the Stuttgart data that were in fact analysed by the three students, Fey, Assunção and Kunz. In Ref. [28*], which is a critical review of current data on the $^{12}\text{C}(\alpha,\gamma)$, the same raw gamma-ray spectra are shown. In 4 slides, numbers 10 to 13, a direct copy of Fey's thesis showing the raw gamma-ray spectra measured below centre of mass energies of 1.7 MeV are shown. We attach to this reply these 4 slides shown in [28*] during the ECT*, Key Reactions in Nuclear Astrophysics conference, February 17, 2025.

In close to 100 raw gamma-ray spectra we do not observe discernible gamma-lines above background. This raises a serious doubt on the validity of the shown angular distributions that were analysed and used to extrapolate the cross section in [30*]. We note that after the presentation at the ECT* M. Gai discussed these matters with R.J. deBoer (and M. Wiescher) and they agreed that the lowest energy angular distributions shown in Fig. 10 of [30*] are not reliable.

This discussion belongs in our paper since one may wonder how data could disagree with a fundamental prediction of quantum mechanics, or as stated by the reviewer, "The issues with previous data sets being inconsistent with unitarity should make future experimenters be aware of this important consistency check". Clearly, the same is true to the previous data shown in [30*]. We feel that our statement "This disturbing observation must be reconciled before such data can be used for extrapolation, but unfortunately they were used by deBoer et al. [30*] to extrapolate the cross section", summarises the view of the reviewer as well as of deBoer who now has second thoughts about using these data. To follow the highlighted Nature editorial guidelines, this has been reworded to "This systematic discrepancy must be understood before such data can be used for extrapolation, an issue that has been overlooked in previous analyses."

The authors limit their comparison to previous experimental data to a plot of the angle integrated cross section and to a plot of the E1/E2 mixing angle. There are two issues with this. First, the total cross section is less sensitive to errors in the angular distribution and thus does not provide an overly useful comparison of the authors' improved differential cross section measurements.

Beside measurements of the differential cross sections, measurements of the total reaction cross section with neutral beams are also a formidable task. It is important to benchmark our data against the world data for total cross sections. This figure also demonstrates a measure of success in the agreement of world data for integrated cross sections measured over the last 51 years. A discussion around comparing our current results and the previously measured

differential cross sections from gamma decay data is presented below in response to another point raised.

The E1/E2 mixing angle comparison is more useful, but it depends on the partial wave decomposition of the cross section procedure, discussed in the text, where further errors, particularly in uncertainty quantification, can occur.

Further text has been included to make the relative contributions to the errors clearer for both the angular distribution and integrated cross section analyses. For the angular distribution analysis, the errors on the partial cross sections and the mixing phase angle are a combination of statistical errors on the fits (dominant), along with the uncertainty on the amount of pure E2 distribution that is added to the fit function to account for the narrow 2^+ resonance at 2.69 MeV (small). The error on the integrated cross sections arises from statistical errors due to counts in each energy bin, uncertainties on the beam intensity, and on the shape of the gamma beam profile. Quantitative statements are included in the revised manuscript, and the full error quota is included in the electronic supplement.

Instead, it would be much more useful for the authors to compare their differential cross sections with other previously published differential cross sections. The differences in the data could shed light on the problems with previous data sets and help to understand if perhaps only some portion of these previous data are in error.

Comparison of our data with previous data could be fraught with issues as we discuss above. Additionally, since the previous historical data were measured in gamma detectors with large solid angle acceptance, it would not be possible to make a meaningful side-by-side comparison of the distributions themselves.

The discussion of usefulness of other data suggested by the reviewer, is beyond the scope of our paper. The goal of this paper and our previous one published by Nature Communications, is to establish the usefulness of our new method to accurately measure angular distributions. Our criterion for identifying accurately measured distributions is the agreement of extracted φ_{12} with elastic scattering data. It has previously been noted that disagreement of φ_{12} leads to substantial differences in the extracted E2/E1 [30*]. This lack of agreement for the existing data is demonstrated in our appendix figure.

The authors state “we conclude that this extrapolation essentially relies on the asymptotic normalisation coefficient (ANC) and, as such, is an indirect determination of the astrophysical cross-section factor.” The phrasing makes it sound as though the authors have discovered something that was not clearly stated in Ref. 30. On the contrary, this assumption is emphasized in a bullet point in a section that lists the assumptions of that work.

We thank the reviewer for highlighting the wording of our statement. We altered the relevant sentence as below:

As pointed out by the authors of ref. [33], this extrapolation essentially relies on the asymptotic normalisation coefficient (ANC) and, as such, is an indirect determination of the astrophysical cross-section.

Could the manuscript be shortened to aid communication of the most significant findings?

I think some of the introduction could be shortened. The authors go into significant detail on the motivation for their measurement, which has been covered by them elsewhere.

The introduction has been condensed in line with the suggestions of the reviewers and the editor.

Have they been fair in their treatment of previous literature?

The authors compare their angle integrated data to other measurements, but the main point of these experiments is to improve on angular distribution measurements. The authors say that comparing with theory or fits is beyond the scope of their work, but they could provide more comparisons with other previously measured angular distribution measurements made at similar energies.

In this paper we concentrate on improving the quality of the data, which we claim so far has not been good enough to engage in theoretical fits. As noted in an earlier response, because of the different experimental techniques used, directly comparing the angular distributions themselves would not be meaningful.

Is the statistical analysis of the data sound?

The analysis seems to be quite thorough. However, what seems to be missing is both the experimental data and a detailed discussion of the uncertainties. For instance, the data seems accurate but not as precise as some past measurements. If they were more precise it would really help to improve the authors' case for these types of measurements. It is unclear what the limiting factors are that prevent more precise measurements.

We agree with the reviewer that better precision would help. We are now engaged in a new measurement to do just that. This is yet another formidable task that requires long beam time allocations that will only become possible once our method is fully established as we present here. We are working on an extended schedule, an effort that spans many years, if not decades, but still shorter than the 51 years since the seminal work of Dyer and Barnes.

Additional sections of text have been added throughout the paper and the methods section to make relative contributions to the sources of experimental uncertainty clearer. These are further shown in the electronic supplement.

Should the authors be asked to provide further data or methodological information to help others replicate their work? (Such data might include source code for modelling studies, detailed protocols or mathematical derivations).

There is something of an inconsistency in the reported data. Figure 2 indicates that there were measurements at 10 energies, but only four angular distributions are shown. What is the reason for this?

Since we reconstruct the energy of each individual event, the measured integrated cross sections could be finely split across the width of the γ beam. However, statistics are generally too low to split up the measured angular distributions in this way, while extracting parameters with reasonable statistical errors. Hence, we analysed one angular distribution per beam at the effective energy. This limitation (purely statistical) does not apply to the angle integrated cross section.

Some additional comments:

1) In the introduction, “Stellar Evolution theory” seems like it could use a reference.

A reference to Rolfs and Rodney, *Cauldrons in the Cosmos* was added [1].

2) A reference should be given for the current uncertainty in the 3a process.

A reference to two of the latest measurements of the radiative width of the Hoyle state have been added [2,3].

3) “... which is not anywhere near the current precision.” What is the current precision? References should be given.

This statement was removed.

4) The E1/E2 mixing angle is not a directly measurable quantity. Please rephrase.

We have clarified in the manuscript that the E1/E2 mixing angle is extracted from the fit to equation 3.

5) “prepared in 2006 by Professor Wolfgang Hammer – senior leader of the Stuttgart group – on the occasion of his retirement”. This seems like superfluous information that is not appropriate for a scientific publication.

We changed this sentence to read: “Prepared by Professor Wolfgang Hammer – former PI of the Stuttgart group”.

6) The center of mass energies listed in Table I do not seem to match the gamma-ray energies labels of Fig. 3. Maybe there is just some rounding. It seems like the gamma-ray energies should also be included in Table I since this is a photodisintegration experiment.

In this work, four gamma ray energies were used. Four angular distributions were analysed, corresponding to each gamma beam energy. For this, the effective centre of mass energies corresponding to each gamma beam are quoted (table II), which include the effects of the γ beam energy profile and the energy variation of the cross section. The centre of mass energies quoted for the integrated cross section values (table I) are reconstructed from the momenta of the outgoing particles in the TPC, on an event-by-event basis. These can therefore be split finer into smaller energy bins. The energies shown in table I therefore do not correspond to specific incident gamma energies.

7) The figures all label the data from this work as “Haverson et al.”. It would seem clearly to label them as “this work” or something similar.

We prefer to refer to ourselves explicitly, since figures get copied from publications and showed by speakers at conferences. As such, in our figure, credit is given explicitly to the work of others but also to the authors of “our work”.

8) The authors state “These initial measurements serve as a strong impetus to continue these challenging measurements to lower energies to allow for accurate extrapolations of the E1 and E2 cross sections to the relevant stellar conditions, without relying on indirect methods.” While it is understood that the extrapolation is beyond the scope of this work, it would greatly strengthen the authors’ assertion that these lower energy measurements are needed by performing some sensitivity studies of extrapolations using estimates of the cross sections they think they can obtain in a separate work.

We are currently engaged in improving current data using our new method. We would like to concentrate on developing and establishing our method and think that such a discussion is not yet warranted.

9) Table III, which I assume would contain the actual differential cross section data, seems to be missing. A detailed accounting of the uncertainties is missing.

The reference to table III was added by mistake and has been removed. All angular distributions and integrated cross section data are provided in the electronic supplement attached to the resubmission. As noted above, additional sections of text have been added throughout the paper and the methods section to make the sources of experimental uncertainty clearer. These are further given in the electronic supplement.

Reviewer 2

The manuscript of Haverson et al. under consideration presents a follow-up paper to Smith et al., Nature Communications 12, 5920 (2021). The 2021 paper presented a new method to address very important experimental uncertainties for nuclear astrophysics associated with the $^{12}\text{C}(\alpha,\gamma)^{16}\text{O}$ reaction rate. The method introduced in 2021 employs gamma beams and a time projection chamber to measure the time-inverse $^{16}\text{O}(\gamma,\alpha)^{12}\text{C}$ reaction, providing some unique advantages over existing methods. The 2021 paper presented a plot of the E1-E2 mixing phase angles measured using the new method demonstrating good agreement with theoretical predictions of quantum mechanics.

The manuscript of Haverson et al. presents new measurements of the same reaction over the same energy range using the same method with some improvements. For example, the fill gas of the TPC has been changed to reduce a significant background and a new technique has been introduced to circumvent the intrinsic energy spread of the gamma beam. The manuscript culminates in a plot of the E1-E2 mixing phase angles similar to the one from the 2021 paper, with modest improvement. The authors elaborate about three importance of the agreement between data and theory in this plot as a quality check of $^{12}\text{C}(\alpha,\gamma)^{16}\text{O}$ data and emphasize that previous data sets from other groups haven't demonstrated agreement.

The astrophysical problem being pursued is a very important one. I believe that this is a nice experimental method, which has been demonstrated to be improved somewhat in the present manuscript. However, I do not believe that there is sufficient technical advancement or novel scientific impact in this manuscript beyond the content of 2021 paper (or content that could have been added to the 2021 paper at the time without the need for additional acquisition of data) to recommend publication in Communications Physics.

We thank the reviewer for their consideration of the manuscript and the summary above. We deem the current paper as a critical step towards future lower energy measurements, which goes beyond our scoping paper of 2021. A great deal of experimental R&D went into getting the TPC to operate effectively with an N_2O gas mixture, which is very challenging. However, as a result, we can be sure that the currently presented data are background free. Agreement with the 2021 data set (in the regions of energy where statistics in the older data set were high enough to extract the ϕ_{12} parameter) tells us something incredibly important – that our earlier methods of ^{12}C removal are indeed valid and do not bias the results. This fact had led to the decision to run future experiments using gamma beams and the new Warsaw TPC at HI γ S using CO_2 gas, which is a much more favourable target from a TPC operation perspective.

We also deem our new ability to measure the centre of mass energy event-by-event as a major step in making the photodissociation technique a leading player in this field of research. It allows us to significantly circumvent the broad energy distribution of the HI γ S gamma beam. Although this method could technically have been feasible with the earlier 2021 data set, it is only presented now as a result of several years of effort in the development of our reconstruction algorithms.

Reviewer 3

The paper "Precision $^{12}\text{C}(\alpha, \gamma)$ Angular Distribution Measurements: First Significant Physics Result with a Time Projection Chamber Operating in Gamma Beams" presents the measurement of the differential cross sections (angular distributions at four energies) of the reaction $^{16}\text{O}(\gamma, \alpha)^{12}\text{C}$ in a time projection chamber using N_2O gas as a target. This was specifically done in the region of a 1- resonance in the $^{12}\text{C}(\alpha, \gamma)$ reaction around $E_{\text{cm}} = 2.4$ MeV, where previous measurements delivered inconclusive results. In addition, total cross sections were extracted over the resonance. This work is a continuation of an existing campaign to develop alternative methods to measure this elusive nuclear reaction; modifications to the setup eliminated previously observed beam induced background.

The work lays out strong evidence for the feasibility of this method and is certainly a good motivation to continue improvements and further measurements with gamma beams and time projection chambers - for example the uncertainty in the two low energy points of the angular distribution should be improved to more strongly constrain the model. As such it is of interest to the broader nuclear astrophysics community (and more) and in principal worthy of publication. The used method and the description of setup and analysis is sound but there are some issues with the introduction, the organisation of the text and the references that need to be addresses before I can suggest acceptance in Nature Communications Physics. Following are my comments.

1) The introduction rests heavily on the recent publication by JWST on the observation of the compact object in the center of SN 1987A. Besides the citation being only to the press release by JWST and not the actual article in Science, which should be corrected, there appears to be some misunderstanding of the statements in said paper: in the present manuscript it is stressed that JWST ruled out a black hole compact object from their new observations, hence giving lead to a necessary revision of the mass cutoff. This is not what is written in the JWST article. Instead, JWST at the beginning of their discussion rules out a black hole simply due to mass considerations and then works only with various neutron star models. Nowhere due they rule out a black hole from the new observations. This point does not diminish the importance of cross section measurements of $^{12}\text{C}(\alpha, \gamma)$, it is after all still one of the most important nuclear reactions in astrophysics, but it appears somewhat odd that the present text relies so heavily on what appears to be some misunderstanding. This can easily be corrected though.

We thank the reviewer for their feedback on this section of the introduction. This section has now been reworded on lines 34 to 48. The reference to the full article published in Science has been added [8], along with another relevant reference [9]. We have made sure that the new text more carefully reflects the explanation from [8], namely that "*Photoionization models show that the line ratios and velocities can be explained by ionizing radiation from a neutron star illuminating gas from the inner parts of the exploded star*", as stated by the editor's summary of [8]. This explanation is in agreement with an earlier observation of a neutrino burst from this region, with a duration indicating the presence of a neutron star [9]. Since the James Webb

observations may provide some insight into the equation of state of nuclear matter, which affects the fate of Type II supernovae, we believe this discussion belongs in our introduction.

2) The organisation of the text is a bit confusing, with the discussion section being in the middle of the text, followed by the methods section. Unless this is a requirement by the journal it seems more natural and easier to follow if the methods section accompanies the presentation of the results and the manuscript concludes with the discussion.

Nature Communications Physics structures their articles in this way.

3) While the authors very nicely describe the analysis in the methods section, a better discussion of systematic uncertainties would be helpful, also in order to judge possible further improvements and intrinsic limits of the method.

Additional sections of text have been added throughout the paper and the methods section to make relative contributions to the sources of experimental uncertainty clearer. These are further shown in the electronic supplement.

4) The references are using a mixture of different styles, something First Name, Last Name, sometimes the other way around. Fully spelled out first names are chosen for some references while abbreviated ones for others. The titles are included or are omitted, without discernible reason for either choice. In case the titles are included, the separator between the last author and the title changes from reference to reference or is not present at all. Lastly, in reference [31] the name of an author is clearly spelled wrong.

We thank the reviewer for highlighting these inconsistencies. The formatting of the references has now been standardised and spelling mistakes were corrected.

The manuscript presents new measurements of the differential cross section of the ground state transition of the $^{12}\text{C}(\text{a},\text{g})^{16}\text{O}$ reaction at energies relevant for nuclear astrophysics application. The measurements were performed in a novel way, using the time reverse reaction. The experimental data are shown to be consistent with unitarity constraints imposed by elastic scattering data and basic quantum mechanical principles, which is not the case for several past data sets. These promising first results, demonstrating the capability of the experimental technique, are presented.

I have used the journal's referee guide to make my comments on the manuscript. They are given below.

The results are novel?

The paper presents new differential cross sections for the photodisintegration of ^{16}O , for which there is very little previous data.

The paper provides strong evidence for its conclusions

The paper has no conclusion section, which seems to be an odd oversight. Section IV, labeled "Discussion" seems to essentially be this however.

The data are technical and sound

They appear to be of very good quality.

The manuscript is important to scientists in the specific sub-field of physics.

Yes, the $^{12}\text{C}(\text{a},\text{g})$ reaction is one of the most important reactions for the field of nuclear astrophysics

In general, to be acceptable, a paper should represent an advance in understanding which may influence thinking in the field. Please see the full scope of the journal for more details.

The paper sheds light on inconsistencies in past data sets and continues to pave the way forward for more accurate measurements.

What are the major claims of the paper?

The authors claim to have measured new experimental data that are more consistent with the unitarity condition of quantum mechanics for this reaction. Several past data sets were not. They make these measurements using photodisintegration, which is a novel technique.

Are the claims novel? If not, please identify the major papers that compromise novelty.

The method is novel.

Will the paper be of interest to others in the field?

The paper is of interest to the field.

Will the paper influence thinking in the field?

The success of the experimental method will make others consider this method of experimentation. The issues with previous data sets being inconsistent with unitarity should make future experimenters be aware of this important consistency check.

Are the claims convincing? If not, what further evidence is needed?

The claims of the paper that their data and that additional experimental data that are consistent with unitarity are needed is convincing.

Are there other experiments that would strengthen the paper further? How much would they improve it, and how difficult are they likely to be?

Yes, experiments with improved precision would greatly strengthen the authors' arguments. It is unclear if they are feasible at this time.

Are the claims appropriately discussed in the context of previous literature?

For the most part, but there are a few places where there are some issues.

In particular, the authors cite their own work as motivation, which is not even a published work (reference 28). This seems odd considering there are other works that are published also clearly provide this motivation, for example, the review by Ref. 30.

The authors limit their comparison to previous experimental data to a plot of the angle integrated cross section and to a plot of the E1/E2 mixing angle. There are two issues with this. First, the total cross section is less sensitive to errors in the angular distribution and thus does not provide an overly useful comparison of the authors' improved differential cross section measurements. The E1/E2 mixing angle comparison is more useful, but it depends on the partial wave decomposition of the cross section procedure, discussed in the text, where further errors, particularly in uncertainty quantification, can occur. Instead, it would be much more useful for the authors to compare their differential cross sections with other previously published differential cross sections. The differences in the data could shed light on the problems with previous data sets and help to understand if perhaps only some portion of these previous data are in error.

The authors state “we conclude that this extrapolation essentially relies on the asymptotic normalisation coefficient (ANC) and, as such, is an indirect determination of the astrophysical cross-section factor.” The phrasing makes it sound as though the authors have discovered something that was not clearly stated in Ref. 30. On the contrary, this assumption is emphasized in a bullet point in a section that lists the assumptions of that work.

If the manuscript is unacceptable in its present form, does the study seem sufficiently promising that the authors should be encouraged to consider a resubmission in the future?

Is the manuscript clearly written? If not, how could it be made more accessible?

The manuscript is clearly written.

Could the manuscript be shortened to aid communication of the most significant findings?

I think some of the introduction could be shortened. The authors go into significant detail on the motivation for their measurement, which has been covered by them elsewhere.

Have the authors done themselves justice without overselling their claims?

I think so.

Have they been fair in their treatment of previous literature?

The authors compare their angle integrated data to other measurements, but the main point of these experiments is to improve on angular distribution measurements. The authors say that comparing with theory or fits is beyond the scope of their work, but they could provide more comparisons with other previously measured angular distribution measurements made at similar energies.

Have they provided sufficient methodological detail that the experiments could be reproduced?

The paper does a very good job of explaining experimental procedures.

Is the statistical analysis of the data sound?

The analysis seems to be quite thorough. However, what seems to be missing is both the experimental data and a detailed discussion of the uncertainties. For instance, the data seems accurate but not as precise as some past measurements. If they were more precise

it would really help to improve the authors' case for these types of measurements. It is unclear what the limiting factors are that prevent more precise measurements.

Should the authors be asked to provide further data or methodological information to help others replicate their work? (Such data might include source code for modelling studies, detailed protocols or mathematical derivations).

There is something of an inconsistency in the reported data. Figure 2 indicates that there were measurements at 10 energies, but only four angular distributions are shown. What is the reason for this?

Are there any special ethical concerns arising from the use of animals or human subjects?

No

Some additional comments:

- 1) In the introduction, "Stellar Evolution theory" seems like it could use a reference.
- 2) A reference should be given for the current uncertainty in the 3a process.
- 3) "... which is not anywhere near the current precision." What is the current precision? References should be given.
- 4) The E1/E2 mixing angle is not a directly measurable quantity. Please rephrase.
- 5) "prepared in 2006 by Professor Wolfgang Hammer – senior leader of the Stuttgart group – on the occasion of his retirement". This seems like superfluous information that is not appropriate for a scientific publication.
- 6) The center of mass energies listed in Table I do not seem to match the gamma-ray energies labels of Fig. 3. Maybe there is just some rounding. It seems like the gamma-ray energies should also be included in Table I since this is a photodisintegration experiment.
- 7) The figures all label the data from this work as "Haverson et al.". It would seem clearly to label them as "this work" or something similar.
- 8) The authors state "These initial measurements serve as a strong impetus to continue these challenging measurements to lower energies to allow for accurate extrapolations of the E1 and E2 cross sections to the relevant stellar conditions, without relying on indirect methods." While it is understood that the extrapolation is beyond the scope of this work, it would greatly strengthen the authors' assertion that these lower energy measurements are needed by performing some sensitivity

studies of extrapolations using estimates of the cross sections they think they can obtain in a separate work.

- 9) Table III, which I assume would contain the actual differential cross section data, seems to be missing. A detailed accounting of the uncertainties is missing.